# EQUIVARIANT HETEROGENEOUS GRAPH NETWORKS

## ABSTRACT

Many real-world datasets include multiple distinct types of entities and relations, and so they are naturally best represented by heterogeneous graphs. However, the most common forms of neural networks operating on graphs either assume that their input graphs are homogeneous, or they convert heterogeneous graphs into homogeneous ones, losing valuable information in the process. Any neural network that acts on graph data should be equivariant or invariant to permutations of nodes, but this is complicated when there are multiple distinct node and edge types. With this as motivation, we design graph neural networks that are composed of linear layers that are maximally expressive while being equivariant only to permutations of nodes within each type. We demonstrate their effectiveness on heterogeneous graph node classification and link prediction benchmarks.

## 1 INTRODUCTION

Many real-world datasets and problems can be modelled as sets of objects with different relationships between them, and so graphs are a natural choice for representing these problems. Common examples include modelling interactions between users in a social network, properties of molecules, or modelling connections between entities in a knowledge base.

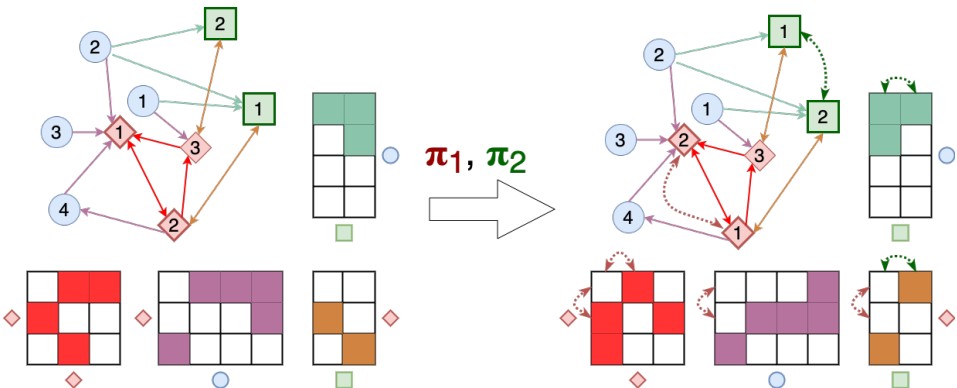

Figure 1: Left: An example heterogeneous graph, with 3 node types and 4 edge types. ◇ represents AUTHORS, ○ represents PUBLICATIONS, and □ represents VENUES. This heterogeneous graph can be represented as a set of adjacency matrices, one for each edge type. In this paper, we design linear mappings between each adjacency matrix, allowing us to learn, for example, how AUTHOR-colleagues-with-AUTHOR relationships and PUBLICATION-published-at-VENUE relationships can influence AUTHOR-associated-with-VENUE relationships.
Right: The effect of applying separate permutations $\pi_1$ and $\pi_2$ to the AUTHOR and VENUE nodes. While the graph itself is unaffected, with nodes simply relabelled, the adjacency matrices are modified. We characterize linear maps that are equivariant to such permutations as pooling and broadcasting operations.

Graph neural networks (GNNs) have become a popular technique for node and graph-level property predictions. These models have mostly focused on standard homogeneous graphs, wherein all nodes and edges are treated the same, with any differences encoded as feature vectors. However,

in practical application settings, data is often complex and multi-typed, necessitating the use of heterogeneous graphs, where nodes and edges can be of different types, with potentially completely different semantics. Typical ways to apply GNNs to heterogeneous networks involve preprocessing techniques such as encoding node and edge types into feature vectors or collapsing heterogeneous networks into homogeneous ones by replacing paths along multiple different edge types with single edges. These techniques reduce the structural information available for any network to learn from and often require domain knowledge and hand-engineered features.

In this paper, we design a neural network that can operate directly on entire heterogeneous graphs while fully respecting the independence and relationships between different node and edge types. We model a heterogeneous graph as a collection of node-node adjacency matrices, one for each edge type, and create mappings from each edge type to every other edge type. For example, in a heterogeneous network that includes PUBLICATIONS, AUTHORS, and VENUES, and the relationships between these entities, our model can learn how PUBLICATION-published-at-VENUE relationships may influence AUTHOR-associated-with-VENUE relationships by constructing a linear mapping between their adjacency matrices; see Fig. 1.

A key property of any neural network that operates on graphs is that they must be *invariant* or *equivariant* to permutations of nodes. That is to say, if a graph is represented by an adjacency matrix $\boldsymbol{A} \in \mathbb{R}^{n \times n}$, for any permutation matrix $\pi \in \{0, 1\}^{n \times n}$, a neural network $f : \mathbb{R}^{n \times n} \to \mathbb{R}^{n \times n}$ must have the property that $f(\pi \boldsymbol{A} \pi^\top) = \pi f(\boldsymbol{A}) \pi^\top$ (equivariance), or when making graph-level predictions with $f : \mathbb{R}^{n \times n} \to \mathbb{R}$, we require invariance $f(\pi \boldsymbol{A} \pi^\top) = f(\boldsymbol{A})$.

For heterogeneous graphs, this invariance or equivariance constraint is to permutations within each node type. We identify all linear operations that map one adjacency matrix to another while maintaining permutation equivariance within each separate node type. By combining these operations, we are able to construct maximally expressive linear equivariant layers that can then be stacked together to produce a heterogeneous graph neural network. We create two different architectures, and apply them to two common heterogeneous graph tasks: node classification, and link prediction. Finally, we extend our treatment to the general case of relationships involving hyperedges between more than two node types, providing a general prescription of how to efficiently implement linear layers that act on heterogeneous hypergraphs.

## 2 RELATED WORK

### 2.1 HETEROGENEOUS GRAPH NETWORKS

The ubiquity of complex multi-typed data in real-world problems has caused heterogeneous graph learning to attract a lot of attention in applied settings. Heterogeneous graph networks have been applied to such diverse tasks as text classification (Linmei et al., 2019), disease diagnosis (Wang et al., 2021), and malicious account detection (Liu et al., 2018).

The majority of heterogeneous graph learning techniques rely on *meta-paths*: sequences of different node and edge types (Sun & Han, 2012; Shi et al., 2017). For example, in a citation network with AUTHORS, PAPERS, AND VENUES, the "path" AUTHOR – PUBLICATION – VENUE – PUBLICATION – AUTHOR represents one meta-path between two AUTHORS that have published at the same venue. These meta-paths are usually hand-designed, requiring domain knowledge.

Heterogeneous graph learning techniques can be broadly classified into either "shallow" embedding models, or "deep" neural models (Dong et al., 2020; Yang et al., 2020). Shallow methods (such as Dong et al. (2017); Tang et al. (2015); Fu et al. (2017)) aggregate node attributes using techniques such as random walks over different edge types, in order to obtain structure-preserving embeddings for each node, which are then passed on to other machine learning models for downstream tasks. These are limited to transductive settings.

Deep methods extend conventional GNNs, but learn parameters or embeddings specific to each node or edge type; see Wu et al. (2020) for a survey of homogeneous GNNs. Examples include R-GCN (Schlichtkrull et al., 2018) which extends GCN by learning edge-specific weight matrices, Heterogeneous Graph Attention Network (HAN, Wang et al., 2019) and Metapath Aggregated Graph Neural Network (MAGNN; Fu et al., 2020), which extend graph attention to attend over different meta-paths. Some methods, such as Heterogeneous Graph Transformer (HGT, Hu et al., 2020)

and Graph Transformer Network (GTN, Yun et al., 2019) can automatically discover what meta-paths are worth using, but even then they are not able to capture as much information as if they were to directly use the full heterogeneous graph. For two recent surveys of heterogeneous graph representation learning techniques, see Yang et al. (2020) and Dong et al. (2020).

Lv et al. (2021) recently called into question whether most heterogeneous graphs neural networks are able to properly exploit the information provided by node and edge types. They show that under fair comparisons, they are often outperformed by conventional graph neural networks that simply ignore node and edge type information, such as GCN (Kipf & Welling, 2016) and GAT (Veličković et al., 2017).

## 2.2 Equivariant and Invariant Learning

A fundamental property that all graph neural networks share is an *equivariance* or *invariance* to node permutations. The general requirement that neural networks must be equivariant and invariant to certain symmetries of their input data has been a very active area of research and has motivated models for a diverse set of data types. Examples include models equivariant to the translational symmetries of images (LeCun et al., 1989), the permutation symmetry of sets (Zaheer et al., 2017), and symmetry to 3D rotations on a sphere (Cohen et al., 2018). As an inductive bias, equivariance of the network to a set of transformations is intuitively equivalent to having seen all such transformations of each training data-point. Since the set of such permutation is exponentially large for graph data, equivariance is essential for any graph neural network.

Several works directly seek the set of operations with this property to use them as building blocks in graph neural networks. Of particular relevance is the work of Kondor et al. (2018), which introduces permutation equivariant operations that can be applied to tensor representations of graphs, and Maron et al. (2018), which characterizes a basis for equivariant linear operations on tensor representations of graphs and hypergraphs. Our work extends this work to the more general case of heterogeneous graphs, where we are presented with a set of different edge types. Furthermore, while they characterise the set of equivariant linear bases, their analysis does not give a practical algorithm since such large matrices that form the linear bases are too large to store in memory for any large graph. While Maron et al. (2018) give an efficient implementation based on pooling and broadcasting for standard homogeneous graphs, they do not provide a general implementation for arbitrary node and edge types. Albooyeh et al. (2019) give a pooling and broadcasting view of operations for hyper-graphs and incidence structures of other geometric entities. However, all such structures have a single node type that is assumed exchangeable. Some other related works that have a symmetry-based approach to GNNs include Maron et al. (2019); de Haan et al. (2020); Azizian & Lelarge (2020).

## 3 Our Model

### 3.1 Notation

A heterogeneous graph $\mathcal{G} = \langle \mathbb{D}, \mathbb{R}, \mathbb{V}, \mathbb{X} \rangle$ is a four tuple, where $\mathbb{D} = \{1, \dots, d\}$ is the set of node types. For each node type $i \in \mathbb{D}$ we have a set of $n_i$ nodes $\mathbb{V}_i = \{v_1, \dots, v_{n_i}\}$. We use $\mathbb{V} = \bigcup_i \mathbb{V}_i$ for the set of all $n = \sum_{i \in \mathbb{D}} n_i$ nodes. $\mathbb{R} = \{r_1, \dots, r_\ell\}$ is the set of edge types, where $r = \langle \overline{r}, \underline{r} \rangle$ for $\overline{r}, \underline{r} \in \mathbb{D}$ is a pair of node types that appear in the edge type r. $\mathbb{X}$ is a set of node adjacency matrices $\mathbb{X} = \{\boldsymbol{X}^r \in \mathbb{R}^{n_{\overline{r}} \times n_{\underline{r}}} \mid r \in \mathbb{R}\}$, one for each edge type $r \in \mathbb{R}$. Such matrices can represent both node and edge attributes using diagonal and off-diagonal elements respectively. For simplicity, we initially assume scalar node and edge attributes, later we show generalization of this to vectors using multiple *channels*. In the definition above, $\mathbb{D}, \mathbb{R}$, and $\mathbb{V}$ contain the blueprint of the heterogeneous graph, while $\mathbb{X}$ contains the actual data.

### 3.2 Equivariance for Heterogeneous Graphs

Given the heterogeneous graph $\mathcal{G}$ our goal is to identify all *equivariant* linear operators that map the set of matrices $\mathbb{X} = \{\boldsymbol{X}_1, \dots, \boldsymbol{X}_\ell\}$ to another set of matrices $\mathbb{Y} = \{\boldsymbol{Y}_1, \dots, \boldsymbol{Y}_\ell\}$ of the same form. For this, it is sufficient to identify all such maps from one edge type to another

$\mathrm{L}^{\mathrm{r}\to\mathrm{r}'} : \mathbb{R}^{n_{\overline{r}}\times n_{\underline{r}}} \to \mathbb{R}^{n_{\overline{r}'}\times n_{\underline{r}'}}$. The overall equivariant map $\mathrm{L}^{\mathbb{R}\to\mathbb{R}}$ can be built from the collection $\mathrm{L}^{\mathrm{r}_1\to\mathrm{r}_1}, \mathrm{L}^{\mathrm{r}_1\to\mathrm{r}_2}, \dots, \mathrm{L}^{\mathrm{r}_1\to\mathrm{r}_\ell}, \dots, \mathrm{L}^{\mathrm{r}_\ell\to\mathrm{r}_\ell}$.

The equivariance condition on the linear operator $\mathrm{L}^{\mathrm{r}\to\mathrm{r}'}$ ensures that any permutation of the input nodes of the same type leads to the same permutation of the nodes in the output for that node type. Let $\pi_i \in S(n_i)$ be a permutation matrix acting on $n_i$ nodes of type $i$. Equivariance constraint requires

$$\mathrm{L}^{\mathrm{r}\to\mathrm{r}'}\big(\pi_{\overline{r}}\boldsymbol{X}^{\mathrm{r}}\pi_{\underline{r}}^{\top}\big) = \pi_{\overline{r}'}\,\mathrm{L}^{\mathrm{r}\to\mathrm{r}'}\big(\boldsymbol{X}^{\mathrm{r}}\big)\pi_{\underline{r}'}^{\top} \quad \forall \pi_{\overline{r}}, \pi_{\underline{r}}, \pi_{\overline{r}'}, \pi_{\underline{r}} \in S(n_{\overline{r}}) \times S(n_{\underline{r}}) \times S(n_{\overline{r}}') \times S(n_{\underline{r}}') \quad (1)$$

where the permutation matrices correspond to two pairs of node types that appear in the input (r) and output edge types (r').

As also observed in related contexts (Kondor et al., 2018; Albooyeh et al., 2019) such linear operators often involve pooling and broadcasting over input and output matrices. Our plan is to enumerate all such operations and prove that these are indeed the only linear operations with the desired equivariance property Eq. (1).

**Example 1.** *To build an intuition for these operations, consider two relations between* $\langle\text{AUTHOR}, \text{VENUE}\rangle$ *and* $\langle\text{PUBLICATION}, \text{AUTHOR}\rangle$*. Let* $\mathrm{r} = \langle 1, 2\rangle$ *denote the former and* $\mathrm{r}' = \langle 3, 1\rangle$ *be the latter, noting that these two edge types have a node type in common. The desired linear map* $\mathrm{L}^{\mathrm{r}\to\mathrm{r}'}$ *should be equivariant to independent permutation of* AUTHOR *nodes,* PUBLICATION *nodes and* VENUE *nodes in our graph. The results that follow this example show that any equivariant* $\mathrm{L}^{\mathrm{r}\to\mathrm{r}'}$ *has the following form:*

$$\mathrm{L}^{\mathrm{r}\to\mathrm{r}'}(\boldsymbol{X}) = w_1\big(\boldsymbol{X}\mathbf{1}_{n_{\underline{r}}}\mathbf{1}_{n_{\overline{r}'}}\big)^{\top} + w_2\mathbf{1}_{n_{\overline{r}'}}\big(\mathbf{1}_{n_{\overline{r}}}^{\top}\boldsymbol{X}\mathbf{1}_{n_{\underline{r}}}\big)\mathbf{1}_{n_{\underline{r}'}}^{\top} \quad (2)$$

*where* $w_1, w_2 \in \mathbb{R}$ *are arbitrary weights and* $\mathbf{1}_n$ *is the identity vector of length* $n$*. Here, following Zaheer et al. (2017) we are performing pooling and broadcasting operations using multiplication by identity vectors. The first operation* $\big(\boldsymbol{X}\mathbf{1}_{n_{\underline{r}}}\mathbf{1}_{n_{\overline{r}'}}\big)^{\top}$ *pools over the columns of* $\boldsymbol{X}$ *(i.e.,* VENUES*), and broadcasts the resulting column vector to create a* $n_{\underline{r}'} \times n_{\overline{r}'}$ *matrix which is then transposed to match the dimensions of the target edge type. We can think of the pooling operation above as collecting edge attributes from all the* VENUES *that are adjacent to each* AUTHOR*. Similarly, the broadcasting operation disperses this pooled information over all the* PUBLICATION *nodes adjacent to each* AUTHOR*. This example shows that an equivariant linear map is able to propagate relevant information across different edge types (within a single layer of a deep network).*

### 3.3 CHARACTERIZING EQUIVARIANT LINEAR MAPS

The question is how to identify all equivariant linear operations for a given pair of edge types $\mathrm{r}, \mathrm{r}'$? In addition to the pooling, broadcasting, and transpose operation used in the example above, we need one additional operation, namely $\mathrm{diag}$. We overload this operation so that for a square matrix $\mathrm{diag} : \mathbb{R}^{n\times n} \to \mathbb{R}^n$ extracts the diagonal, and for a vector input $\mathrm{diag} : \mathbb{R}^n \to \mathbb{R}^{n\times n}$ the output is a square matrix with that vector on its diagonal – this means $\mathrm{diag}(\mathrm{diag}(\boldsymbol{x})) = \boldsymbol{x}$ and $\mathrm{diag}(\mathrm{diag}(\boldsymbol{X})) = \boldsymbol{X}\odot\boldsymbol{I}$ (where $\odot$ is the Hadamard product and $\boldsymbol{I}$ is an identity matrix).

The idea is to create all possible combinations of the linear operations above that take us from a $n_{\overline{r}} \times n_{\underline{r}}$ matrix to a $n_{\overline{r}'} \times n_{r'}$ matrix. These operations vary based on the equality of some of these dimension – for example if $\overline{r} = \underline{r}$ then the operation $\mathrm{diag}(\boldsymbol{X})$ is well-defined, and otherwise it is not feasible. To help with this enumeration, we break any such linear operation into parts:

**Contraction operations** These include pooling over the rows, columns, both rows and columns, extraction of diagonal and pooling over the diagonal, as well as the identity operation. The result could be a scalar, a vector, or a matrix. Below, we use $z, \boldsymbol{z}$, and $\boldsymbol{Z}$ to denote these intermediate products, and identify the condition under which we can perform each of these contraction operations:

| Operation | | Condition |
|---|---|---|
| 1. Identity operation | $\boldsymbol{Z}^{\langle \overline{r}, \underline{r} \rangle} = \boldsymbol{X}^{\langle \overline{r}, \underline{r} \rangle}$ | - |
| 2. Pooling over columns | $\boldsymbol{z}^{\overline{r}} = \boldsymbol{X}^{\langle \overline{r}, \underline{r} \rangle} \boldsymbol{1}_{n_{\underline{r}}}$ | - |
| 3. Pooling over rows | $\boldsymbol{z}^{\underline{r}} = \boldsymbol{X}^{\langle \overline{r}, \underline{r} \rangle^\top} \boldsymbol{1}_{n_{\overline{r}}}$ | - |
| 4. Pooling over rows and columns | $z = \boldsymbol{1}_{n_{\overline{r}}}^\top \boldsymbol{X}^{\langle \overline{r}, \underline{r} \rangle} \boldsymbol{1}_{n_{\underline{r}}}$ | - |
| 5. Extracting the diagonal | $\boldsymbol{z}^{\overline{r}} = \mathrm{diag}(\boldsymbol{X}^{\langle \overline{r}, \underline{r} \rangle})$ | $\overline{r} = \underline{r}$ |
| 6. Pooling the diagonal | $z = \mathrm{diag}(\boldsymbol{X}^{\langle \overline{r}, \underline{r} \rangle})^\top \boldsymbol{1}_{n_{\overline{r}}}$ | $\overline{r} = \underline{r}$ |

**Expansion operations** These operations expand the intermediate value to produce the target matrix. The operations include broadcasting over rows, columns, both rows and columns, diagonal placement, diagonal broadcasting, as well as the identity operation and matrix transpose.

| | | |
|---|---|---|
| 1. Identity operation | $\boldsymbol{Y}^{\langle \overline{r}', \underline{r}' \rangle} = \boldsymbol{Z}^{\langle \overline{r}, \underline{r} \rangle}$ | $\overline{r}' = \overline{r}, \underline{r}' = \underline{r}$ |
| 2. Transpose | $\boldsymbol{Y}^{\langle \overline{r}', \underline{r}' \rangle} = \boldsymbol{Z}^{\langle \overline{r}, \underline{r} \rangle^\top}$ | $\overline{r}' = \underline{r}, \underline{r}' = \overline{r}$ |
| 3,4. Broadcasting over columns | $\boldsymbol{Y}^{\langle \overline{r}', \underline{r}' \rangle} = \boldsymbol{z}^{\overline{r}} \boldsymbol{1}_{n_{\underline{r}'}}^\top$ | $\overline{r} = \overline{r}'$ |
| | $\boldsymbol{Y}^{\langle \overline{r}', \underline{r}' \rangle} = \boldsymbol{z}^{\underline{r}} \boldsymbol{1}_{n_{\underline{r}'}}^\top$ | $\underline{r} = \overline{r}'$ |
| 5,6. Broadcasting over rows | $\boldsymbol{Y}^{\langle \overline{r}', \underline{r}' \rangle} = \boldsymbol{1}_{n_{\overline{r}'}} \boldsymbol{z}^{\overline{r}^\top}$ | $\overline{r} = \underline{r}'$ |
| | $\boldsymbol{Y}^{\langle \overline{r}', \underline{r}' \rangle} = \boldsymbol{1}_{n_{\overline{r}'}} \boldsymbol{z}^{\overline{r}^\top}$ | $\underline{r} = \underline{r}'$ |
| 7. Broadcast over rows and cols | $\boldsymbol{Y}^{\langle \overline{r}', \underline{r}' \rangle} = \boldsymbol{1}_{n_{\overline{r}'}} z \boldsymbol{1}_{n_{\underline{r}'}}^\top$ | - |
| 8,9. Placing the diagonal | $\boldsymbol{Y}^{\langle \overline{r}', \underline{r}' \rangle} = \mathrm{diag}(\boldsymbol{z}^{\overline{r}})$ | $\overline{r}' = \underline{r}' = \overline{r}$ |
| | $\boldsymbol{Y}^{\langle \overline{r}', \underline{r}' \rangle} = \mathrm{diag}(\boldsymbol{z}^{\underline{r}})$ | $\overline{r}' = \underline{r}' = \underline{r}$ |
| 10. Broadcasting over the diagonal | $\boldsymbol{Y}^{\langle \overline{r}', \underline{r}' \rangle} = \mathrm{diag}(z \boldsymbol{1}_{n_{\overline{r}'}})$ | $\overline{r}' = \underline{r}'$ |

**Theorem 3.1.** *Given two edge types* $\mathrm{r}, \mathrm{r}'$, *all the linear maps* $\mathrm{L}^{\mathrm{r} \to \mathrm{r}'} : \mathbb{R}^{n_{\overline{r}} \times n_{\underline{r}}} \to \mathbb{R}^{n_{\overline{r}'} \times n_{\underline{r}'}}$ *that satisfy the equivariance condition of Eq.* (1) *are produced using the contraction and expansion operations above.*

*Proof.* This is a special case of Theorem 6.1. □

### 3.4 THE FEEDFORWARD LAYER

Now that we have enumerated all possibilities for permutation equivariant linear mappings $\mathrm{L}^{\mathrm{r} \to \mathrm{r}'}$, we can combine them to form a linear layer that acts on a set of adjacency matrices:

$$\mathrm{L}^{\mathbb{R} \to \mathbb{R}}(\mathbb{X}) = \left\{ \sum_{\boldsymbol{X}^{\mathrm{r}} \in \mathbb{X}} \sum_j w_j \, \mathrm{L}_j^{\mathrm{r} \to \mathrm{r}'}(\boldsymbol{X}^{\mathrm{r}}) \mid \mathrm{r}' \in \mathbb{R} \right\} \tag{3}$$

Here, $j$ indexes all valid combinations of contraction and expansion operations, and $w_j \in \mathbb{R}$ is a weight for that combination that may be learned. In practice, both the inner and outer sum can be replaced by any permutation invariant aggregation function, such as taking the maximum or taking the mean.

#### 3.4.1 MULTIPLE CHANNELS

We can extend the above definitions to include edge feature vectors in a straightforward way, if we instead replace the matrix $\boldsymbol{X}^{\mathrm{r}}$ with the tensor $\mathbf{X}^{\mathrm{r}} \in \mathbb{R}^{n_{\overline{r}} \times n_{\underline{r}} \times f}$ where $f$ is some feature dimension. Now, instead of having scalar weights $w_j$ in Eq. (3), we have a collection of weight matrices $\boldsymbol{W}_j \in \mathbb{R}^{f' \times f}$. Each equivariant layer can then specify the number of feature dimensions in their input and output.

#### 3.4.2 SPARSE IMPLEMENTATION

In practice, graphs tend to be very sparse, making it impractical to deal with full adjacency matrices. We instead represent an adjacency matrix with $m$ nonzero entries as a tuple $\boldsymbol{X} = \langle \boldsymbol{I}, \boldsymbol{V} \rangle$, where

$I \in \mathbb{Z}^{2 \times m}$ are the indices of nonzero values, and $V \in \mathbb{R}^{m \times f}$ are the nonzero values. Each of the contraction and expansion operations of Section 3.3 for a mapping $L^{r \to r'}$ can be implemented using sparse operations with a space complexity of $\mathcal{O}(m + m' + n_{\overline{r}} + n_{\underline{r}} + n'_{\overline{r}} + n'_{\underline{r}})$ and a time complexity of $\mathcal{O}\left((m + m') \log(m + m') + n_{\overline{r}} + n_{\underline{r}} + n'_{\overline{r}} + n'_{\underline{r}}\right)$.

The $\log$-factor is because the Identity and Transpose expansion operations require that we match the nonzero indices of the input and output adjacency matrices, an operation that involves sorting, giving it a time complexity of $\mathcal{O}((m + m') \log(m + m'))$. However, this only needs to be computed once for a given input and output sparsity mask, rather than for every pass through a layer.

This means that the feedforward layer effectively has a linear complexity in the number of nodes and edges of the graph, making it efficient for large datasets. However, inducing the sparsity on the output of the feedforward layer should be seen as an non-linear operation. Since permutation of node types also permutes the sparsity patterns, this non-linear operation is equivariant.

### 3.4.3 Encoding and Decoding Layers

It is also useful to have matrix-to-vector encoding and vector-to-matrix decoding layers. For example, an encoding layer can take in a set of adjacency matrices and output embeddings for each node of each type, while a decoding layer can take in node embeddings and output values for each possible edge, which may be used for link prediction. Maximally expressive linear encoding (decoding) layers can be constructed by combining all valid contraction (expansion) operations that are of the form $\mathbb{R}^{n_{\overline{r}} \times n_{\underline{r}}} \to \mathbb{R}^{n_r'}$ ($\mathbb{R}^{n_r} \to \mathbb{R}^{n_{\overline{r}'} \times n_{\underline{r}'}}$). Their explicit form is left to Appendix C.

## 4 Tasks and Architectures

Our heterogeneous graph layers, just like regular linear layers in a multilayer perceptron, can be stacked together and alternated with nonlinear activations to form a variety of neural network architectures. We describe here two conventional graph learning tasks, and what architectures we designed for them.

### 4.1 Node Classification

For the task of node classification, we are provided with a heterogeneous graph, where a subset of nodes of one target type are labelled. We are tasked with predicting the labels of the other nodes of the target type.

We use an architecture consisting of a stack of equivariant heterogeneous graph layers separated by nonlinear activation functions. We apply batch normalization over the nonzero entries for each of the matrices at each layer, and we use channel-wise dropout at each layer to prevent overfitting. Following our stack of equivariant heterogeneous graph layers, we add an encoding layer. This encoding layer can either be used to directly predict classes for each node, or they can be used to get an embedding vector for each node which is then fed into a conventional linear classifier to get predictions. The network is trained using a negative log loss over labels.

### 4.2 Link Prediction

For the task of node classification, we are provided with a heterogeneous graph where a subset of edges of one target type have been removed. Given a set of candidate edges, half of which are real and half of which have one node with the tail node randomly replaced with another 2-hop neighbour, we are tasked with assigning a confidence score to each potential edge.

To accomplish this task, we use an autoencoder architecture. We create a stack of equivariant heterogeneous graph layers separated by nonlinearities, followed by an encoding layer that produces node embeddings for each node of each node type. This makes up the encoding module of our autoencoder. These node embeddings are then passed into a decoding layer, producing matrices for each edge type. These matrices are passed through another stack of equivariant heterogeneous graph layers, outputting a confidence score for each potential edge for the target edge type. The

neural network is trained using binary cross-entropy loss over a 1:1 mix of positive samples of real edges of the target edge type, and randomly sampled negative edges.

## 5 EVALUATION

We evaluate our architectures using the recently created Heterogeneous Graph Benchmark (HGB) (Lv et al., 2021), which gives a set of standardized datasets and training/test splits in node classification and link prediction. To prevent any possible test set leakage, test set labels are withheld, and evaluation metrics are obtained by submitting predictions to the HGB website[1]. We make comparisons against the heterogeneous graph neural networks Simple-HGN (Lv et al., 2021), RGCN (Schlichtkrull et al., 2018), HAN (Wang et al., 2019), GTN (Yun et al., 2019), RSHN (Zhu et al., 2019), HetGNN (Zhang et al., 2019), MAGNN (Fu et al., 2020), HetSANN (Hong et al., 2020), HGT (Hu et al., 2020), GCN (Kipf & Welling, 2016), and GAT (Veličković et al., 2017). All evaluation scores listed here are taken from Lv et al. (2021). Details on the specific hyperparameters searched over and used for each dataset, are included in Appendix B. A link to our anonymized code is included here: `https://github.com/equivariant-hgn/equivariant_hgn`.

Table 1: Characteristics of each of the datasets tested (Lv et al., 2021).

| Node Classification | Nodes | Node Types | Edges | Edge Types | Node Attributes | Target | Classes |
|---|---|---|---|---|---|---|---|
| **DBLP** | 26,128 | 4 | 239,566 | 6 | Yes | Author | 4 |
| **IMDB** | 21,420 | 4 | 86,642 | 6 | Yes | Movie | 5 |
| **ACM** | 10,942 | 4 | 547,872 | 8 | Yes | Paper | 3 |
| **FREEBASE** | 180,098 | 8 | 1,057,688 | 36 | No | Book | 7 |
| Link Prediction | | | | | | Target | |
| **AMAZON** | 10,099 | 1 | 148,659 | 2 | Yes | Product-product | |
| **LASTFM** | 20,612 | 3 | 141,521 | 3 | No | user-artist | |
| **PUBMED** | 63,109 | 4 | 244,986 | 10 | Yes | disease-disease | |

### 5.1 NODE CLASSIFICATION

We look at four datasets: DBLP, IMDB, ACM, and Freebase. DBLP, ACM, and Freebase are multi-class classification tasks, and IMDB is a multi-label task. All datasets except for Freebase additionally include node attributes, and 24% of target nodes labels are used for training, 6% for validation, and 70% for testing. Further information is included in Table 1, which is adapted directly from Lv et al. (2021). The task is evaluated using the metrics of Micro-F1 and Macro-F1 scores (F1 scores that have been averaged over all nodes and all labels respectively).

A comparison between our results and competing methods is shown in Table 2. It can be seen that our method generally performs comparably with other top methods, and yields higher performance than the state of the art for two particular metrics.

### 5.2 LINK PREDICTION

We look at three datasets: Amazon, LastFM, and PubMed. Amazon and PubMed additionally include node attributes, and PubMed also includes edge attributes. While our method is easily able to incorporate edge attributes, we ignored them in the case of PubMed in order to make a fair comparison with other methods that do not use them. For each dataset, 81% of edges of the target edge type are used for training, 9% are used for validation, and 10% are withheld for the test set. Further information on these datasets is included in Table 1.

Edge predictions are evaluated using two metrics: The area under the Receiver Operating Characteristic curve (ROC-AUC), and the Mean Reciprocal Rank (MRR). The ROC-AUC score evaluates

---

[1]https://www.biendata.xyz/hgb/

Table 2: Comparison of our method on the node classification task.

| Method | DBLP | | IMDB | |
|---|---|---|---|---|
| | Macro-F1 | Micro-F1 | Macro-F1 | Micro-F1 |
| Simple-HGN | **94.01±0.24** | **94.46±0.22** | 62.05±1.36 | **67.36±1.36** |
| RGCN | 91.52±0.50 | 92.07±0.50 | 58.85 ± 0.26 | 62.05±0.15 |
| HAN | 91.67±0.49 | 92.05±0.62 | 57.74±0.96 | 64.63±0.58 |
| GTN | 93.52±0.55 | 93.97±0.54 | 60.47±0.98 | 65.14±0.45 |
| RSHN | 93.34±0.58 | 93.81±0.55 | 59.85±3.21 | 64.22±1.03 |
| HetGNN | 91.76±0.43 | 92.33±0.41 | 48.25±0.67 | 51.16±0.65 |
| MAGNN | 93.28±0.51 | 93.76±0.45 | 56.49±3.20 | 64.67±1.67 |
| HetSANN | 78.55±2.42 | 80.56±1.50 | 49.47±1.21 | 57.68±0.44 |
| HGT | 93.01±0.23 | 93.49±0.25 | 63.00±1.19 | 67.20±0.57 |
| GCN | 90.84±0.32 | 91.47±0.34 | 57.88±1.18 | 64.82±0.64 |
| GAT | 93.83±0.27 | 93.39±0.30 | 58.94±1.35 | 64.86±0.43 |
| **Equivariant HGN (ours)** | 92.79±0.33 | 93.29±0.3 | **63.15±1.06** | 66.67 ±0.92 |
| Method | ACM | | FREEBASE | |
| | Macro-F1 | Micro-F1 | Macro-F1 | Micro-F1 |
| Simple-HGN | **93.42±0.44** | **93.35±0.45** | 47.72±1.48 | **66.29±0.45** |
| RGCN | 91.55±0.74 | 91.41±0.75 | 46.78±0.77 | 58.33±1.57 |
| HAN | 90.89±0.43 | 90.79±0.43 | 21.31±1.68 | 54.77±1.40 |
| GTN | 91.31±0.70 | 91.20±0.71 | - | - |
| RSHN | 90.50±1.51 | 90.32±1.54 | - | - |
| HetGNN | 85.91±0.25 | 86.05±0.25 | - | - |
| MAGNN | 90.88±0.64 | 90.77±0.65 | - | - |
| HetSANN | 90.02±0.35 | 89.91±0.37 | - | - |
| HGT | 91.12±0.76 | 91.00±0.76 | 29.28±2.52 | 60.51±1.16 |
| GCN | 92.17±0.24 | 92.12±0.23 | 27.84±3.13 | 60.23±0.92 |
| GAT | 92.26±0.94 | 92.19±0.93 | 40.74±2.58 | 65.26±0.80 |
| **Equivariant HGN (ours)** | 92.26±0.44 | 92.17 ±0.45 | **48.35±1.57** | 63.42±0.29 |

the model's ability to discriminate between real and fake edges over different sensitivity thresholds. The MRR score evaluates the ability's model to rank real candidate edges higher than false edges.

A comparison between our results and competing methods is shown in Table 3. Our method performs comparably to other leading methods on the LastFM benchmark, outcompetes all other methods on the Amazon benchmark, and for the PubMed benchmark it outcompetes all other methods by a very large margin. The PubMed dataset contains many node and edge types, and we believe that our model is able to perform so well by learning the interactions between each of these relationships.[2]

Table 3: Comparison of our method on the link prediction task.

| Method | AMAZON | | LASTFM | | PUBMED | |
|---|---|---|---|---|---|---|
| | ROC AUC | MRR | ROC AUC | MRR | ROC AUC | MRR |
| Simple-HGN | 93.40±0.62 | 96.94±0.29 | **67.59±0.23** | **90.81±0.32** | 83.39±0.39 | 92.07±0.26 |
| RGCN | 86.34±0.28 | 93.92±0.16 | 57.21±0.09 | 77.68±0.17 | 78.29±0.18 | 90.26±0.24 |
| GATNE | 77.39±0.50 | 92.04±0.36 | 66.87±0.16 | 85.93±0.63 | 63.39±0.65 | 80.05±0.22 |
| HetGNN | 77.74±0.24 | 91.79±0.03 | 62.09±0.01 | 83.56±0.14 | 73.63±0.01 | 84.00±0.04 |
| MAGNN | - | - | 56.81±0.05 | 72.93±0.59 | - | - |
| HGT | 88.26±2.06 | 93.87±0.65 | 54.99±0.28 | 74.96±1.46 | 80.12±0.93 | 90.85±0.33 |
| GCN | 92.84±0.34 | 97.05±0.12 | 59.17±0.31 | 79.38±0.65 | 80.48±0.81 | 90.99±0.56 |
| GAT | 91.65±0.80 | 96.58±0.26 | 58.56±0.66 | 77.04±2.11 | 78.05±1.77 | 90.02±0.53 |
| **Equivariant HGN (ours)** | **96.75±0.16** | **97.78±0.15** | 60.94±0.36 | 82.36±0.71 | **99.89±0.01** | **99.86±0.03** |

---

[2]While our results for the PubMed dataset are suspiciously strong, because the test-set is withheld by the curators of the dataset and also used by other methods in this table, we see no obvious reason to doubt the performance of our model. However, we have communicated this to the curators in case there is any issue with data processing and split.

## 6 HETEROGENEOUS HYPERGRAPHS

In this section, we extend the theory of Section 3 to hypergraphs, where a *hyperedge type* $r = \langle r(1), \ldots, r(m) \rangle$ relates $m$ node types. What was previously a set of matrices is now a set of tensors $\mathbb{X} = \{\mathbf{X}^r \in \mathbb{R}^{n_{r(1)} \times \ldots \times n_{r(m)}} \mid r \in \mathbb{R}\}$.

Similar to Eq. (3) in heterogeneous graphs, the linear map for this setup also decomposes into blocks, and it is sufficient to identify the form of equivariant linear maps $L^{r \to r'} : \mathbf{X}^r \mapsto \mathbf{Y}^{r'}$. To characterize all such equivariant linear maps, we consider all combinations of pool and broadcast operations with extraction and placement of hyper-diagonals. To facilitate this, we introduce the following notation for these four operations:

**Pooling** $\mathrm{pool}_{\mathbb{P}}(\mathbf{X})$ pools over all the node indices $\mathbb{P}$ of its input – *e.g.*, $\mathrm{pool}_{\{1\}} X^{\langle 2,1 \rangle} = \mathbf{1}_{n_2}^{\top} X^{\langle 2,1 \rangle}$
and $\mathrm{pool}_{\{1,2\}} X^{\langle 2,1 \rangle} = \mathbf{1}_{n_2}^{\top} X^{\langle 2,1 \rangle} \mathbf{1}_{n_1}$.

**Broadcasting** $\mathrm{broadcast}_{r'}(\mathbf{Z}^r)$ broadcasts the input tensor $\mathbf{Z}^r$ into a tensor corresponding to hyperedge type $r'$. For example $\mathrm{broadcast}_{\langle 2,1 \rangle} x^{\langle 2 \rangle} = x \mathbf{1}_{n_1}^{\top}$. This requires the elements of $r$ to appear in $r'$ – more accurately $\mathrm{multi\text{-}set}(r) \subseteq \mathrm{multi\text{-}set}(r')$, where $\mathrm{multi\text{-}set}$ returns its input tuple as a multi-set.

**Extracting a hyper-diagonal** A hyper-diagonal generalizes the notion of diagonal of a matrix. Given a tensor $\mathbf{X}^{\langle r(1), \ldots, r(m) \rangle}$, a hyper-diagonal is identified by a partitioning $\mathbb{H}$ of the set $\{1, \ldots, m\}$ where $r(i) = r(j)$ whenever $i, j$ are in the same partition. With this definition of hyperdiagonal, the extraction operation $\mathrm{extract\text{-}diag}_{\mathbb{H}}(\mathbb{X}^r)$ reproduces the effect of $\mathrm{diag}$ operation for a matrix by simply using $\mathbb{H} = \{\{1, 2\}\}$. In the extreme case where $\mathbb{H} = \{\{1\}, \ldots, \{m\}\}$ identifies the entire input tensor as the hyper-diagonal, this operation becomes the identity operation.

**Placing a hyper-diagonal** $\mathrm{place\text{-}diag}_{r', \mathbb{K}}(\mathbf{Z}^r)$ places the tensor $\mathbf{Z}^r$ over the hyper-diagonal of a tensor $\mathbf{Y}^{r'}$ identified by the partition $\mathbb{K}$.

We can write any equivariant linear operation: $L^{r \to r'} : \mathbb{R}^{n_{r(1)} \times \ldots \times n_{r(m)}} \to \mathbb{R}^{n_{r'(1)} \times \ldots \times n_{r'(m')}}$ as a linear combination of different "compatible" choices for these four operations:

$$\mathbf{Y}^{r'} = L^{r \to r'}(\mathbf{X}^r) = \sum w_{\mathbb{H}, \mathbb{P}, r'', \mathbb{K}} \, \mathrm{place\text{-}diag}_{r, \mathbb{K}} \left( \mathrm{broadcast}_{r''} \left( \mathrm{pool}_{\mathbb{P}} \left( \mathrm{extract\text{-}diag}_{\mathbb{H}}(\mathbf{X}^r) \right) \right) \right) \quad (4)$$

This composition of operations is first extracting a hyper-diagonal using $\mathbb{H}$, pooling over a subset of indices identified by $\mathbb{P}$, broadcasting some of these dimensions and permuting them according to $r''$, and finally placing the resulting tensor over a hyper-diagonal of the output tensor, identified by $\mathbb{K}$. Next, we show that these are the only operations needed to create an equivariant linear layer for homogeneous hyper-graphs.

**Theorem 6.1.** *All equivariant linear maps* $L^{r \to r'} : \mathbb{R}^{n_{r(1)} \times \ldots \times n_{r(m)}} \to \mathbb{R}^{n_{r'(1)} \times \ldots \times n_{r'(m')}}$ *between two hyperedge types in a hypergraph are of the form Eq. (4).*

*Proof.* The full proof of this theorem is included in Appendix A. □

## 7 CONCLUSIONS

By enumerating all equivariant linear operations that can be applied to the data structure of heterogeneous graphs, we have demonstrated how they may be combined to create effective heterogeneous graph neural networks. We demonstrated their effectiveness in the task of link prediction, their competitiveness for node classification, and give an efficient implementation that can be used for any collection of arbitrary interactions between nodes of different types. Further work is possible to evaluate the effectiveness of equivariant heterogeneous graph networks on the task of heterogeneous graph classification (a task where the expressive abilities of a network are very important), and learning of heterogeneous hypergraphs (an underexplored area). While we only created two relatively simple architectures using these equivariant layers, in principle we could create much more sophisticated models in the same way that sophisticated neural networks are based on simple linear layers.

ACKNOWLEDGMENTS

[Removed for anonymity]

REPRODUCIBILITY STATEMENT

The authors have made efforts to ensure reproducibility of our experimental results and clarity in our theoretical contributions. An anonymous repository containing our code is included here: `https://github.com/equivariant-hgn/equivariant_hgn`. The experimental setup follows from the publicly accessible Heterogeneous Graph Benchmark (Lv et al., 2021), with architectural details, including hyperparameters searched over and used for each experiment, included in Appendix B. Our theoretical contributions are all based on Theorem 6.1, for which we have provided a proof in Appendix A.

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

# A  PROOF OF THEOREM 6.1

*Proof.* It is easy to see that all the four operations used in Eq. (4) are equivariant. In order to show that these operations exhaust all possibilities, we count the number of compatible choices for $\mathbb{H}, \mathbb{P}, r'', \mathbb{K}$ for a given input/output pair of edge types r and r'. Through this counting, we arrive at the same number of operations as what is given by (Maron et al., 2018)'s Theorem 3, where the maximality is also established. A related problem for equivariant linear maps for incidence networks appears in Albooyeh et al. (2019), and the following proof is inspired by the combinatorial counting arguments in that paper.

Let $\left\{ {p \atop q} \right\}$ be the number of ways we can partition a set of size $p$ into $q$ non-empty partitions. This is also known as the *Stirling partition number*. Moreover, we use $\kappa(r, i)$ for $i \in \mathbb{D}$ to denote the number of occurrences of node type $i$ in edge type r. Now we claim that the total number of compatible choices for $\mathbb{H}, \mathbb{P}, r'', \mathbb{K}$ is given by

$$\prod_{i=1}^{d} \sum_{m,m'=1}^{\min\{\kappa(r,i),\kappa(r',i)\}} \left\{ {\kappa(r,i) \atop m} \right\} \left\{ {\kappa(r',i) \atop m'} \right\} \sum_{l=0}^{\min\{m,m'\}} \binom{m}{l}\binom{m'}{l} l! \tag{5}$$

Because our operations for each node type are independent, the first product is over all possible node types. In the next summation, we only consider the occurrences of node type $i$, and partition these in both r and r' into $m$ and $m'$ non-empty partitions respectively. The subsequent Stirling numbers count the number of ways in which we can produce these partitions. In the inner summation, we select $l$ of these $m$ and $m'$ partitions to match them against each other. The number of such possible choices is given by the number of ways we can select $l$ out of $m$ and $m'$ partitions (given by the Binomial coefficients), times all the possible pairings over these $l$ partitions for the matching purpose ($l!$).

Now that we know what the expression above is counting, let us explain the connection to Eq. (4). The intuitive motivation is that we want to enumerate all possible pairings of outputs of extract-diag with inputs of place-diag. In Eq. (6), the number $m$ represents the order of the output tensor of extract-diag with node type $i$, and $m'$ represents the order of the input tensor to place-diag. The Stirling numbers are counting the number of different hyper-diagonals of the input and output tensors $\mathbf{X}^r$ and $\mathbf{Y}^{r'}$ respectively. Once we identify $l$ of these partitions on hyper-diagonals to match, the remaining dimensions from the input hyper-diagonal are pooled, while we broadcast over those of the output hyper-diagonal.

Now we write the combinatorial expression of Eq. (4) in an alternate form:

$$\prod_{i=1}^{d} \sum_{m,m'=1}^{\min\{\kappa(r,i),\kappa(r',i)\}} \left\{ {\kappa(r,i) \atop m} \right\} \left\{ {\kappa(r',i) \atop m'} \right\} \sum_{l=0}^{\min\{m,m'\}} \binom{m}{l}\binom{m'}{l} l! \tag{6}$$

$$= \prod_{i=1}^{d} \sum_{l=0}^{\min\{\kappa(r,i),\kappa(r',i)\}} \sum_{m=l}^{\kappa(r,i)} \sum_{m'=l}^{\kappa(r,i)} \left( \binom{m}{l} \left\{ {\kappa(r,i) \atop m} \right\} \right) \left( \binom{m'}{l} \left\{ {\kappa(r',i) \atop m'} \right\} \right) l! \tag{7}$$

$$= \prod_{i=1}^{d} Bell(\kappa(r,i) + \kappa(r',i)) \tag{8}$$

where in Eq. (7), we simply re-arrange the summations in Eq. (6). In arriving at Eq. (8) from Eq. (7) we use a combinatorial argument: recall that the Bell number $Bell(k)$ is the number of different ways we can partition $k$ objects into non-empty partitions. To see why Eq. (7) is counting the same number of partitions of $\kappa(r,i) + \kappa(r',i)$ objects, first partition each of these two sets into any number $m, m' \geq l$ partitions. Next, merge $l$ of those partitions from the first and second set in all possible ways to create a partitioning of $\kappa(r,i) + \kappa(r',i)$ into $m + m' - l$ partitions. It is easy to see that this procedure does not produce the same partitioning twice and all different partitions of $\kappa(r,i) + \kappa(r',i)$ are produced in this way. This last expression is what appears in Maron et al. (2018) in Theorem 3. The argument above shows that the number of different ways we can perform Eq. (4) is equal to this Bell number and therefore all equivariant linear maps of interest have this form. □

# B   HYPERPARAMETERS

For both the link prediction and the node classification task, we used the Adam Optimizer with weight decay. We also optionally apply a fully connected layer to the graph node attributes before passing it on to the rest of our network. For each dataset, we ran sweeps on a range of hyperparameters, evaluating their performance against a held-out validation set. The hyperparameters used and the range we tested over are included in Table 4. The sets of hyperparameter values that yielded the best performance on the validation set for each dataset are included in Table 5.

Table 4: Hyperparameters tested for each task.

| Task | Hyperparameter | Description | Sweep Range |
|---|---|---|---|
| Both | ACT_FN | Nonlinear activation function | ReLU, LeakyReLU, Tanh |
| | DROPOUT | Channel-wise dropout | 0, 0.1, 0.3, 0.5 |
| | LR | Optimizer learning rate | 1e-3, 5e-4, 1e-4 |
| | POOL_OP | Pooling operation used instead of the inner summation in Eq. (3) | mean, max |
| | WEIGHT_DECAY | Optimizer weight decay | 1e-3, 1e-4, 1e-5, 1e-6 |
| | WIDTH | Number of feature dimensions for each equivariant layer | 16, 32, 64 |
| Node Classification | DEPTH | Number of equivariant layers + optional input fully connected layer | 1, 2, 3, 4, 5, 6 |
| | FC_LAYER | Input dimension of optional additional fully connected layer after obtaining node embeddings | 0, 16, 32, 64, 128 |
| | FEATS_TYPE | If True, ignore node features of non-target nodes | True, False |
| | IN_FC_LAYER | If true, the first layer of the network is set to be a fully connected layer instead of an equivariant layer | True, False |
| Link Prediction | DEPTH | Number of equivariant and fully connected layers in both the encoding and decoding modules | 2, 3, 4, 5, 6 |
| | EMBEDDING_DIM | Dimensions of node embeddings | 32, 64, 128 |
| | IN_FC_LAYER | If true, the first layer of the encoding module and the last layer of the decoding module are set to be fully connected layers instead of equivariant layers | True, False |

Table 5: Hyperparameters selected for each dataset for both tasks.

| | Node Classification | | | | Link Prediction | | |
|---|---|---|---|---|---|---|---|
| | **DBLP** | **IMDB** | **ACM** | **FREEBASE** | **AMAZON** | **LASTFM** | **PUBMED** |
| ACT_FN | LeakyReLU | LeakyReLU | ReLU | Tanh | ReLU | ReLU | ReLU |
| DEPTH | 6 | 6 | 3 | 6 | 5 | 4 | 3 |
| DROPOUT | 0 | 0.3 | 0.3 | 0 | 0.1 | 0 | 0 |
| EMBEDDING_DIM | - | - | - | - | 64 | 64 | 32 |
| FC_LAYER | 32 | 128 | 64 | 16 | - | - | - |
| FEATS_TYPE | 1 | 1 | 0 | 1 | - | - | - |
| IN_FC_LAYER | False | False | False | True | True | True | True |
| LR | 0.001 | 0.0001 | 0.001 | 0.0005 | 0.001 | 0.001 | 0.0001 |
| POOL_OP | max | mean | mean | mean | mean | mean | max |
| WEIGHT_DECAY | 0.001 | 0.0001 | 1.00E-05 | 1.00E-06 | 0.001 | 0.0001 | 0.0001 |
| WIDTH | 64 | 64 | 64 | 16 | 128 | 128 | 32 |

## C  ENCODING AND DECODING LAYERS

We define here an equivariant encoding mapping $\mathrm{P}^{\mathrm{r}\to r'} : \mathbb{R}^{n_{\overline{r}}\times n_{\underline{r}}} \to \mathbb{R}^{n'_r}$ and an equivariant decoding mapping $\mathrm{B}^{r\to\mathrm{r}'} : \mathbb{R}^{n_r} \to \mathbb{R}^{n_{\overline{r}'}\times n_{\underline{r}'}}$. The equivariance conditions on these mappings are:

$$\mathrm{P}^{\mathrm{r}\to r'}\big(\pi_{\overline{r}}\boldsymbol{X}^{\mathrm{r}}\pi_{\underline{r}}^{\top}\big) = \pi_{r'}\mathrm{P}^{\mathrm{r}\to r'}\big(\boldsymbol{X}^{\mathrm{r}}\big) \quad \forall \pi_{\overline{r}}, \pi_{\underline{r}}, \pi_{r'} \in S(\overline{r}) \times S(\underline{r}) \times S(r') \tag{9}$$

$$\mathrm{B}^{r\to\mathrm{r}'}\big(\pi_r \boldsymbol{z}^r\big) = \pi_{\overline{r}'}\mathrm{B}^{r\to\mathrm{r}'}\big(\boldsymbol{z}^r\big)\pi_{\underline{r}'}^{\top} \quad \forall \pi_r, \pi_{\overline{r}'}, \pi_{\underline{r}'} \in S(r') \times S(\overline{r}') \times S(\underline{r}') \tag{10}$$

**Theorem C.1.** *Given an edge type* $\mathrm{r}' = \langle \overline{r}', \underline{r}'\rangle$ *and a node type* $\overline{r}$, *all the linear maps* $\mathrm{P}^{\mathrm{r}'\to\overline{r}} :$ $\mathbb{R}^{n_{\overline{r}}\times n_{\underline{r}}} \to \mathbb{R}^{n_{\overline{r}}}$ *that satisfy the equivariance condition of Eq.* (9) *are produced using the valid contractions of the form* $\mathbb{R}^{n_{\overline{r}}\times n_{\underline{r}}} \to \mathbb{R}^{n_{r'}}$ *in Section 3.3. All linear maps* $\mathrm{B}^{r\to\mathrm{r}'} : \mathbb{R}^{n_{\overline{r}}} \to \mathbb{R}^{n_{\overline{r}}\times n_{\underline{r}}}$ *that satisfy the equivariance condition of Eq.* (10) *are produced using the valid expansions of the form* $\mathbb{R}^{n_r}\to\mathbb{R}^{n_{\overline{r}'}\times n_{\underline{r}'}}$ *in Section 3.3.*

*Proof.* As with Theorem 3.1, these are also special cases of Theorem 6.1. $\qquad\square$

As with the standard equivariant layers we've defined, these equivariant mappings can be combined together to form layers:

$$\mathrm{P}^{\mathbb{R}\to\mathbb{D}}(\mathbb{X}) = \left\{ \sum_{\boldsymbol{X}^{\mathrm{r}}\in\mathbb{X}} \sum_{j} \mathrm{P}_j^{\mathrm{r}\to d}(\boldsymbol{X}^{\mathrm{r}}) \mid d' \in \mathbb{D} \right\} \tag{11}$$

$$\mathrm{B}^{\mathbb{D}\to\mathbb{R}}(\mathbb{Z}) = \left\{ \sum_{\boldsymbol{z}^d\in\mathbb{Z}} \sum_{j} \mathrm{B}_j^{d\to\mathrm{r}}(\boldsymbol{z}^d) \mid \mathrm{r} \in \mathbb{R} \right\} \tag{12}$$

where for $\mathrm{P}$, $j$ indexes each valid contraction operation, and for $\mathrm{B}$, $j$ indexes each valid expansion operation.

