# OpenReview forum: "Equivariant Heterogeneous Graph Networks"
_ICLR.cc/2022/Conference — ICLR 2022 Submitted_

### Official Review · Reviewer_Nimp · 2021-10-27

**Correctness:** 3
**Technical Novelty And Significance:** 2
**Empirical Novelty And Significance:** 2
**Recommendation:** 5
**Confidence:** 4

**Main Review:**


Strength:

1. Heterogeneous graph neural networks are common in different applications.
   It is of importance to extend previous permutation-equivariant graph neural
   networks to the heterogeneous setting.

2. Some of the experimental results are very positive, for example, on the PubMed dataset. The method achieves almost perfect link prediction performance.

Weakness:

1.The novelty of the proposed method is marginal. There has already been
previous works that make graph neural network permutation equivalent. Maron et
al. (2018) and Albooyeh et al. (2019) have used equivariant pooling and
broadcasting operations for building equivariant graph neural network models.
In this work, the core idea is very similar, and the contribution is to extend
that idea to multiple types of nodes and edges, which is of marginal novelty
in my opinion.

2. A major concern about the experiments is that many important baselines are
missing. For the compared baselines, none of them are designed for
permutation-equivalent, but there do exist graph neural networks that are
permutation-equivalent. Without comparing with such baselines, it's hard to
judge how much this work has advanced this direction.

3. The performance of the proposed model has highly varying performance on
different tasks and data sets. For example, for link prediction, the model
achieves almost perfect performance on PubMed, but the gaps are much smaller
on the other two datasets. For node classification, the performance of the
model is sometimes even worse than the vanilla graph neural network. Although
the author briefly explained their performance variance on link prediction,
it's much better to give a more thorough and detailed analysis.



**Summary Of The Paper:**

This paper proposes a heterogeneous graph neural network model that is
permutation-equivalent. It achieved permutation-equivalence by defining a set
of linear mapping operations (contraction and expansion) that are permutation
equivariant for of type-specific adjacency matrix. In the experiments, the
model has shown positive results compared with non-equivariant heterogeneous
graph neural networks in link prediction and node classification.


**Summary Of The Review:**

This paper is studying an interesting problem of permutation-equivariant
heterogeneous graph neural network. Some of the experimental results are very
positive. However, the novelty of this paper is marginal, the results are
highly varying, and some important baselines are missing.

---

### Official Review · Reviewer_cyQD · 2021-11-02

**Correctness:** 4
**Technical Novelty And Significance:** 2
**Empirical Novelty And Significance:** 1
**Recommendation:** 5
**Confidence:** 4

**Details Of Ethics Concerns:**

Given that GNNs can be used along with social network to predict user attributes that otherwise the users may want to keep private, I would encourage the authors to write an ethics statement for the same as given in the ICLR guidelines.

**Main Review:**

The paper is written well and easy to follow.

## Strengths:
1. The paper proposes a model and details the various operations under which it maintains the equivariance property.
2. It goes on to show that all linear maps that satisfy the invariance property can be made from the given set of operations.
3. In addition, it also shows that the neural networks are highly expressive.
4. Results for link prediction tasks (particularly for PubMed is pretty impressive).

## Weaknesses:
1. Proposed novelty is marginal in the light of existing prior works.
2. The results are not as impressive for node classification and in link prediction, it falls behind significantly for Last FM.
3. There is no explanation given for these phenomenon. I would encourage the authors to add explanation as to why an expressive GNN with equivariance property is not able to leverage itself better for node classification and on the Last FM dataset.

In light of the above, I am inclined to rate this marginally below acceptance threshold.

**Summary Of The Paper:**

Most Graph Neural Networks (GNNs) are often proposed for homogeneous graphs or convert heterogeneous graphs into homogeneous graph and utilize them. This paper proposes a graph neural network which maintains its expressivity while retaining the equivariance property.

**Summary Of The Review:**

The paper is clearly written and easy to follow. The theoretically analysis is fairly clean and the model, as a whole, is very cleanly motivated. The results on link prediction tasks, particularly on Pubmed, is impressive. However, novelty is marginal in light of existing prior works and the performance for node classification and link prediction on Last FM is quite underwhelming.

In light of the above, I am inclined to rate this marginally below acceptance threshold.

----

Update post response phase:
Most reviewers do agree that this is an interesting and important problem and that the authors have done a good job laying out the various operations with equivariant property. Also, most reviewers also have identified that some of the experiment results are very positive and encouraging as well.

However, limited novelty is a point that most other reviewers have also raised. I also noticed that they requested for additional baseline comparisons which seems like a fair request. Also, the performance of the model doesn't seem to be stable, sometimes its good and in some cases, it much worse and there doesn't seem to be any reasonable explanation for the same. This is another point, that has quite come up in other reviews as well.

In light of this, I am inclined to retain my current assessment of marginally below acceptance threshold.

---

### Official Review · Reviewer_Q6T2 · 2021-11-02

**Correctness:** 4
**Technical Novelty And Significance:** 3
**Empirical Novelty And Significance:** 3
**Recommendation:** 5
**Confidence:** 2

**Main Review:**

Pros:
- The performance is impressive for edge prediction. Not that this is not the case for node prediction. Experiments do not show a statistically significant improvement in the node classification task over H-GCN.
-The authors do a nice job at formalizing the types linear mapping that they allow, and stating the desiderata of GNNs in terms of relative exchangeability of nodes.
Cons:
- The authors do not test their method fully.  It would have been interesting to test the method a little more through artificial datasets, allowing the authors to control the ground truth --- would they have then been able to state why their method was so much more impressive on PubMed? Does it have to do with the sparsity of the graph? Or the number of different edges as they posit? More insights based on the limitations and ideal use case of this method would be useful. As such, the paper does not address any fundamental questions on the deployability of the method.
- The results on PubMed do look a little suspicious --- the authors do say so themselves, so this is not necessarily a con, but more of an axis of reflexion: I wonder how this could be explained. Could it be the case that the network is really overfitting? It would have been interesting to list the complexities (in terms of the number of parameters) of the different methods. The splitting of the network in test and training set is completely artificial in the GNN literature ---  it's all on the same network, so as long as we're doing message-passing, this seems really artificial, and not providing what we want to achieve by splitting the data into training and testing. What we need is an estimate of "out-of-sample" error or risk. I suppose the method here is not inductive, and cannot be applied to a new node, so we're stuck....Unless we try out methods in the spirit of [1], and we subsample neighborhood or subnetworks within our large graph, fit the Neural Network parameters based on empirical loss functions such as [1]? The authors could also have done ablation studies, and perturbation studies (maybe these could give indications towards overfitting?).

[1] Veitch, Victor, et al. "Empirical risk minimization and stochastic gradient descent for relational data." The 22nd International Conference on Artificial Intelligence and Statistics. PMLR, 2019.


**Summary Of The Paper:**

The focus of this paper is in the design of GNNs that are amenable to operate on heterogeneous graphs --- that is, networks where relationships can be of different types. Current methods do not account for these different types of edges, or reduce everything to the same graph --- thereby loosing valuable information.

In this case, the heterogeneous graph is modelled as a collection of node-node adjacency matrices, one for each edge type. The additional desiderata that they add here is that  that the GNNs that they seek must be invariant or equivariant to permutations of nodes (depending on whether we're doing node level or graph level predictions). For heterogeneous graphs, this invariance or equivariance constraint is to permutations within each node type (eg, within authors, or venues). To build a repository of linear operations that verify these properties, the authors simply enumerate them.

**Summary Of The Review:**

Overall, this is an interesting paper, that includes a nice formalization of the problem of defining equivariant operators in GNNs. However, the validation of the proposed method is not convincing enough, as it does not stress-test the method, nor does it provide insights into when the method should be applied with respect to other SOTA approaches.

---

### Official Review · Reviewer_XzQS · 2021-11-02

**Correctness:** 3
**Technical Novelty And Significance:** 2
**Empirical Novelty And Significance:** 2
**Recommendation:** 5
**Confidence:** 4

**Main Review:**

Strength

1 - Propose a new equivariant operation-based model for heterogeneous graph representation learning.

2 - Presentation is clear for me.

Weakness

1 - The novelty of this work is incremental.

2 - Lack discussion of important related works.

3 - Experiment should be improved.

Detailed Review
The novelty of proposed method is incremental for me. This is not the first work to study equivariant graph neural network, discussion of important related work is missing, such as:

E(n) Equivariant graph neural network, ICML 2021

Despite of the difference in equivariant operation design, I do not think this work is novel and the combination of numerous equivariant operations is not a significant contribution.

In addition, the experiments should be improved. First, since this model involves different types of equivariant operations, it is necessary to study the impacts of different operations (ablation study). Second, according to Table 2/3, the performance of proposed model is worse than the best baseline methods for many cases, raising the concern of model effectiveness. Third, the authors mentioned hypergraphs in Section 6, while not experiments support this extended model.

Minor issues also exist, such as typo. For example, the first sentence of section 4.2 should be: for the task of link prediction...

**Summary Of The Paper:**

This paper proposes an equivariant heterogeneous graph neural network model for heterogeneous graph representation learning. The proposed model introduces numerous operations over adjacency matrix and combines them as final aggregation result. Experiments are conducted to show that the proposed method outperforms some baseline methods.

**Summary Of The Review:**

The novelty of this work is incremental. It lacks existing equivariant graph neural network discussion and comparison. In addition, experiments should be improved.

---

### Author Response · Authors · 2021-11-22
**Thank You**

We greatly thank the reviewers for their well-considered and practical feedback. In particular, all reviews noted a lack of sufficient explanations of why our method performs better on some datasets and tasks while underperforming in others. We are investigating this, and will address these concerns in a future re-submission to another venue.

---

### Decision · Program_Chairs · 2022-01-20

**Decision:**

Reject

**Comment:**

This paper proposes a design of GNNs that are amenable to operating on heterogeneous graphs. The proposed model introduces numerous operations over the adjacency matrix and combines them as a final aggregation result. Experiments are conducted to show that the proposed method outperforms some baseline methods.

The submission suffers from, an incremental novelty, missing important references, and unconvinced experiments.

All reviewers tend to reject this submission before and after the rebuttal.